# AGNOSTIC PERSONALIZED FEDERATED LEARNING WITH KERNEL FACTORIZATION

## ABSTRACT

Considering the futuristic scenarios of federated learning at a worldwide scale, it is highly probable that local participants can have their own personalized labels, which might not be compatible with others (even for the same class), and can be also possibly from a variety of multiple domains. Nevertheless, they should be benefited from others while selectively taking helpful knowledge. Toward such extreme scenarios of federated learning, however, most existing approaches are limited in that they often assume: (1) labeling schemes are all synchronized amongst clients; (2) the local data is from the same single dataset (domain). In this sense, we introduce loosely constrained federated learning, namely *Agnostic Personalized Federated Learning (APFL)*, where any clients, regardless of what they have learned with their personalized labels, can collaboratively learn while benefiting each other. We then study two essential challenges of the agnostic personalized federated learning, which are (1) *Label Heterogeneity* where local clients learn from the same single domain but labeling schemes are not synchronized with others and (2) *Domain Heterogeneity* where the clients learn from the different datasets which can be semantically similar or dissimilar to each other. To tackle these problems, we propose our novel method, namely *Similarity Matching and Kernel Factorization (SimFed)*. Our method measures task-level similarity based on locally learned knowledge and matches the relevant ones for personalized knowledge reflection. Furthermore, we factorize our model parameters into two basis vectors and the highly sparse masks to significantly reduce the dimensionlaity of parameter space for alleviating knowledge collapse and information loss when reflecting the heterogeneous knowledge. We extensively validate our method on both single- and multi-domain datasets, showing that our method outperforms the current state-of-the-art federated learning methods.

## 1 INTRODUCTION

*Federated Learning (FL)* aims to learn the universal global model while merging locally-learned knowledge from multiple clients rather than collecting the local data directly to the central server (McMahan et al., 2017; Li et al., 2020b; Wang et al., 2020; Karimireddy et al., 2021; Li et al., 2021). Although such decentralized features of federated learning have various advantages, such as preservation of the data privacy, reduction of computing and data maintenance costs, etc., it at the same time shows significant vulnerability in terms of the local heterogeneity. When the local data is not independent and identically distributed (Non-IID), the locally-learned knowledge might not be mutually helpful depending on the level of distributional shifts, yielding performance degradation of the universal model, which is considered as the critical challenge of federated learning.

One of the successful solutions to such problem is *Personalized Federated Learning (PFL)*, whose main goal is to adapt individual clients to their own local distribution better, rather than learning the single universal model to generalize over all local heterogeneous distributions, which might be sub-optimal (Arivazhagan et al., 2019; Liang et al., 2020; Fallah et al., 2020; Zhang et al., 2021). For example, FedFOMO (Zhang et al., 2021), which is the current state-of-the-art approach, downloads a few set of random other clients' knowledge and then selectively takes only beneficial knowledge that are performing well on the local obejctive, which is different form the conventional federated learning methods that often aggregate all local knowledge. Since such personalized federated learning

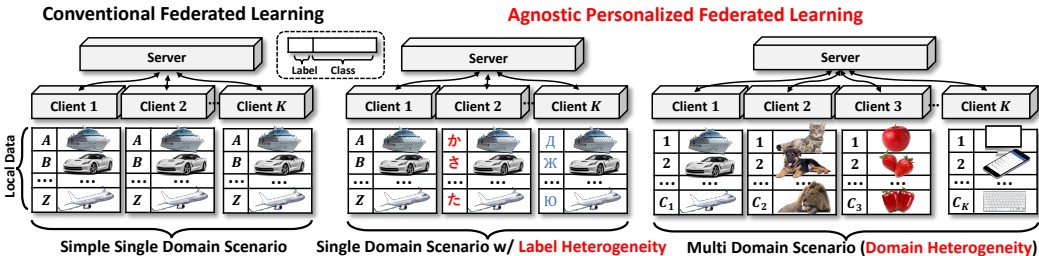

Figure 1: **Overview of the Conventional and Agnostic Personalized Federated Learning:** *Left* illustrates the conventional federated learning. ***Middle*** represents label heterogeneity where labels are not synchronized with others. ***Right*** shows the multi-domain scenario where each local client learns $K$ different datasets.

approaches are client-oriented, they have shown stronger robustness compared to the the traditional federated learning approaches in handling the local heterogeneity at each local device.

Toward further realistic scenarios of federated learning at world-wide scale, however, most existing methods are limited in that they often suppose that (1) target labels are the exactly same across all clients and (2) local data is from a certain single dataset (domain). For the former supposition, all clients share the synchronized labeling schemes in the single domain scenarios (see Figure 1 Left), while it is more natural that labels are slightly different depending on the user environment (see Figure 1 Middle). For example, diverse labels, such as "Cat", "Kitty", or "Neko" (in Japanese), can be used for the single "Cats" class. Furthermore, both index and number of classes can vary per client, such as {1:Cats, 2:Dogs, ..., 10:Horse} for some clients, {1:Horses, 2: Cats, ..., 7: Dogs} for the other clients. We find out that such *Label Heterogeneity* is highly non-trivial problem in that the most current approaches are significantly degenerated (see Table 1).

For the latter assumption, majority of existing methods mainly tackle distributional shifts, e.g. Non-IID, sampled by a certain distribution, e.g. Dirichlet dist., from the same single dataset. However, considering multiple domains is highly essential (see Figure 1 Right), since sharing knowledge even between the heterogeneous domains can be beneficial to each other depending on the semantic similarity or dissimilarity amongst the local domains. For instance, if we have to learn COVID-19 very quickly but with scarce data, utilizing the similar knowledge, e.g. Pneumonia, can be more effective than learning from scratch or irrelevant knowledge, e.g. vehicles. We also find that discriminating which domain knowledge is beneficial to which domain is also very difficult, as the existing methods fail to handle such *Domain Heterogeneity* (see Table 2).

To this end, we introduce an extremely realistic problem of federated learning, namely **Agnostic Personalized Federated Learning (APFL)**, where any clients, regardless of what they have learned with their personalized labels, can collaboratively learn while benefiting each other. We then study two critical challenges of it: (1) **Label Heterogeneity** which labeling schemes are not synchronized for each client; (2) **Domain Heterogeneity** which each client learns non-identical datasets. To tackle the problems, we propose a novel method, namely **Similarity Matching and Kernel Factorization (SimFed)**. Our method measures task-level similarity based on locally learned knowledge and matches the relevant ones for personalized knowledge reflection. Furthermore, we factorize our model parameters into two basis vectors and the highly sparse masks to significantly reduce the dimensionlaity of parameter space for alleviating knowledge collapse and information loss when reflecting the heterogeneous knowledge. We validate our method on both single domain datastes (Permuted IID/Non-IID) and multi-domain datasets (CIFAR-100 20 superclasses) and show our method outperforms the current state-of-the-art approaches on all experiments. In sum, the main contributions of our work are as follow:

- We introduce a realistic problem of **Agnostic Personalized Federated Learning (APFL)**, and discuss two possible critical challenges of it, namely **Label-** and **Domain-Heterogeneity**.

- We propose a novel method to tackle the problems, namely **Similarity Matching and Kernel Factorization (SimFed)**, which measures the task-level similarity and factorizes network parameters to significantly reduce dimensionality for alleviating knowledg collapse when reflecting the heterogeneous knowledge.

- We extensively validate our method in both label- and domain-heterogeneous scenarios and show that our method outperforms the current state-of-the-art baselines.

## 2 RELATED WORK

**Federated Learning**  A variety of algorithms have been proposed for federated learning after firstly introduced by McMahan et al. (2017), but we specifically focus on studies tackling the heterogeneity problems, e.g. Non-IID. FedProx (Li et al., 2020b) intorduces proximal regularization to enforce locally-learned knowledge to be close to the global knowledge. FedMA (Wang et al., 2020) learns the layer-wise permutation when averaging weights, while SCAFFOLD (Karimireddy et al., 2021) utilizes control variates to adjust for the client-drift in the local updates. MOON (Li et al., 2021) performs model-level contrastive learning between global and local models. In this paper, as we are pursuing the task-level heterogeneity, we do not cover the architecture-level heterogeneity (Seo et al., 2020; Zhu et al., 2021; Diao et al., 2021; Shamsian et al., 2021).

**Personalized Federated Learning** aims to improve the individual local clients instead of learning the universal global model via clustering-based approaches (Sattler et al., 2019; Ghosh et al., 2021), the mixture methods (Mansour et al., 2020; Deng et al., 2020; Hanzely & Richtárik, 2021), or meta-learning approach (Fallah et al., 2020). Except for them, FedPer (Arivazhagan et al., 2019) divides local models into base and personalized layers and shares only base layers, while LG-FedAvg (Liang et al., 2020) instead shares the prediction layers with the others and keeps the base layers locally. FedFOMO (Zhang et al., 2021), the current state of the art method, downloads and selectively aggregate other clients' knowledge.

**Learning from Multiple Domains** Beyond learning from the single domain, several recent studies have been proposed to achieve the universal global model well-generalized over multiple domains via the mixture of the client target distributions (Mohri et al., 2019), a distributed multi-task-learning framework (Smith et al., 2018; Li et al., 2020a), domain adaptation for unlabeled data (Peng et al., 2020; Yao et al., 2021). All these prior works, which aim to learn the global model from the multiple domains, however, are orthogonal to our direction, which is to adapt the individual client to their own local domain well that is mainly based on personalized federated learning approaches.

**Other Remarkable Methods** Similarly to our kernel factorization method Jeong et al. (2021); Yoon et al. (2021) decompose model parameters and train them with different objective. The main difference between ours and the prior works is that ours is designed to reduce the dimensionality of parameters into the low rank vectors for alleviating information loss often caused by the conventional coordinate wise aggregation (Wang et al., 2020). Our similarity matching method is similar to (Duan et al., 2021; Sattler et al., 2019) which use client-level similarity. While the methods calculate the raw gradient updates for discriminating relevant clients, our method does not utilize gradient updates directly to calculate similarity between clients. Rather, ours uses efficiently and effectively encoded model representations to measure client-level relevance.

## 3 PROBLEM DEFINITION

We begin with the formal definition of the conventional federated learning and describe *Agnostic Personalized Federated Learning (APFL)*. We then define two essential scenarios for *Label Heterogeneity* and *Domain Heterogeneity*, respectively.

### 3.1 PRELIMINARIES

Our main task is to solve image classification problem using deep neural networks. Let $f_g$ be global model (neural network) at the global server and $\mathcal{F} = \{f_k\}_{k=1}^K$ be a set of $K$ local neural networks. $\mathcal{D} = \{\mathbf{x}_i, y_i\}_{i=1}^N$ be a given dataset, where $K$ is the number of local models, $N$ is the number of instances, $\mathbf{x}_i \in \mathbb{R}^{W \times H \times D}$ is $i$th examples in a size of width $W$, height $H$, and depth $D$, with a corresponding target class $y_i \in \{1, \ldots, C\}$ for the $C$-way multi-class classification problem. The given dataset $\mathcal{D}$ is then disjointly split into $K$ sub-partitions $\mathcal{P}_k = \{\mathbf{x}_{k,i}, y_{k,i}\}_{i=1}^{N_k}$ s.t. $\mathcal{D} = \bigcup_{k=1}^K \mathcal{P}_k$, which are distributed to the corresponding local model $f_k$. Let $R$ be the total number of the communication rounds and $r$ be the current round index. At the first round $r$=1, the global model $f_g$ initialize the global weights $\theta_{f_g}^{(1)}$ and broadcasts $\theta_{f_g}^{(1)}$ to an arbitrary subset of local models that are available for training at round $r$, such that $\mathcal{F}^{(r)} \subset \mathcal{F}$, $|\mathcal{F}^{(r)}| = K^{(r)}$, and $K^{(r)} \leq K$, where $K^{(r)}$ is the number of available local models at round $r$. Then the active local

models $f_k \in \mathcal{F}^{(r)}$ perform local training to minimize loss $\mathcal{L}(\theta_k^{(r)})$ on the corresponding sub-partition $\mathcal{P}_k$ and update their local weight $\theta_k^{(r+1)} \leftarrow \theta_k^{(r)} - \eta \nabla \mathcal{L}(\theta_k^{(r)})$, where $\theta_k^{(r)}$ is local weights for local model $f_k$ at round $r$ and $\mathcal{L}(\cdot)$ is the loss function to minimize locally. When the local training is done, the global model $F$ collects and aggregates the learned weights $\theta_{f_g}^{(r+1)} \leftarrow \frac{N_k}{N} \sum_{i=1}^{K^{(r)}} \theta_k^{(r)}$ and then broadcasts newly updated weights to the local models available at the next round $r+1$. These learning procedures are repeated until the final round $R$, and this entire framework refers to as *Federated Learning*. On the other hand, *Personalized Federated Learning* aims to adapt the individual local models $f_{1:K}$ to their local data distribution $\mathcal{P}_{1:K}$ better rather than achieving the globally universal model $f_g$ over all local distribution, and thus merging local knowledge is not necessarily in a form of $\theta_{f_g}^{(r+1)} \leftarrow \frac{N_k}{N} \sum_{i=1}^{K^{(r)}} \theta_k^{(r)}$ depending on the algorithms.

## 3.2 AGNOSTIC PERSONALIZED FEDERATED LEARNING

*Agnostic Personalized Federated Learning* is personalized federated learning where any local participants from diverse domains with their own personalized labeling schemes can collaboratively learn while benefiting each other. To achieve such objective, following two critical challenges should be tackled: (1) *Label Heterogeneity* and (2) *Domain Heterogeneity*.

**Scenario for Label Heterogeneity**    This scenario assumes that the labeling schemes are not perfectly synchronized across all clients, as described in Section 1 and Figure 1 Middle. Most underlying setting for this scenario is the same as the conventional single-domained setting described in Section 3.1, except that labels are arbitrarily permuted amongst clients. The local data $\mathcal{P}_k$ for the local model $f_k$ is now defined as $\mathcal{P}_k = \{\mathbf{x}_{k,i}, \varphi_k(y_{k,i})\}_{i=1}^{N_k}$, where $\varphi_k(\cdot)$ is a mapping function for local model $f_k$ that maps a given class $y_{k,i}$ with a randomly permuted label $p_{k,i} = \varphi_k(y_{k,i})$. Let the $j$th layer out of $L$ layers in the neural networks of local model $f_k$ be $\ell_k^j$ and the last layer $\ell_k^L$ be a classifier layer. Now that each client has differently permuted labels, the personalized classifiers $\ell_{1:K}^L$ are no longer compatible each other, and thus they are enforced to be remained locally and not undesirably merged for the fairness of this label-heterogeneous scenario.

**Scenario for Domain Heterogeneity**    This scenario presumes that local clients learn on their own entire datasets $\mathcal{D}$, rather than partitions $\mathcal{P}_k$ from the single dataset, as described in Section 1 and Figure 1 Right. In this setting, $K$ disjoint datasets $\mathcal{D}_{1:K}$ are assigned to the $K$ local clients $f_{1:K}$, where $\mathcal{D}_k = \{\mathbf{x}_{k,i}, y_{k,i}\}_{i=1}^{N_k}$ is the entire dataset assigned to the local model $f_k$, where the number of target classes are possibly different, such that $y_{k,i} \in \{1, \ldots, C_k\}$. Between the datasets, there is *no instance-wise and class-wise overlapping*, such that $\varnothing = \bigcap_{k=1}^{K} \mathcal{D}_k$. Similarly to the label-heterogeneous scenario described above, each client has disjoint target classes, and thus the personalized classifiers $\ell_{1:K}^L$ are no longer compatible each other. Hence, they are enforced to be remained locally and not shared each other for the fairness of this domain-heterogeneous scenario.

## 4 SIMFED: SIMILARITY MATCHING AND KERNEL FACTORIZATION

We now elaborate on in-depth description of our novel algorithm for agnostic personalized federated learning, namely *Similarity Matching and Kernel Factorization (SimFed)* in the following subsections. For better understanding, our overall framework is illustrated in Figure 2

### 4.1 SIMILARITY MATCHING

We first define beneficiality of clients as the degree of performance improvement when merging their knowledge. If merging knowledge between certain clients improves individual model's performance (or global model performance), then those participants can be seen as beneficial to each other. If not, participants can be seen as not helpful to each other, which we should avoid merging them.

Zhang et al. (2021) demonstrated that avoiding irrelevant clients improves local performance. They suggested to download few other clients' weights and validate them locally for discriminating helpful models. This is an intuitive approach, but we want a more efficient and effective way to find helpful models. How can we measure task-level similarity in an efficient and effective way? Our assumption

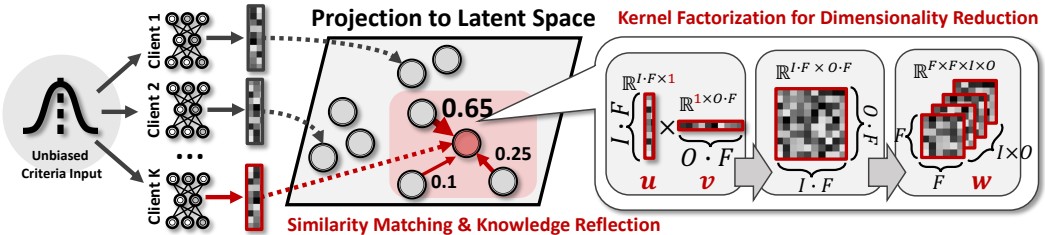

Figure 2: **Illustration of Similarity Matching and Kernel Factorization:** We match relevant clients based on the model embeddings obtained from the criteria input. Then we reflect the relevant knowledge based on the similarity. Our kernel weights are factorized to reduce the dimensionality of parameters

is that we can identify those helpful models if we can recognize task-level similarity, as Yoon et al. (2021) revealed that clients learned on similar tasks improves local performance.

For this, our simple-yet-effective idea is to view the individual local models as respective set encoders that can effectively transform their local data distribution to their own personalized embedding vectors. If we equally feed the same inputs to all encoders, then they will interpret the criteria inputs depending on what they have learned in their own local environments and produce their own personalized transformations, which we can efficiently utilize them as representations of local knowledge to measure the relevance. Formally, the neural networks $f_k$ can be represented as:

$$f_k(\hat{\mathbf{y}}|\mathbf{x}) = P_k(\hat{\mathbf{y}}|E_k(\mathbf{z}|\mathbf{x})), \tag{1}$$

where $f_k$ is a combination of an encoder $E_k(\mathbf{z}|\mathbf{x})$ and a predictor $P_k(\hat{\mathbf{y}}|\mathbf{z})$, where $\mathbf{z} \in \mathbb{R}^d$ is $d$-length transformed latent vector of input data $\mathbf{x}$ and $\hat{\mathbf{y}} \in \mathbb{R}^C$ is a logit for the final prediction. We generate the unbiased criteria input $\mathbf{x}_{\mathcal{N}} \in \mathbb{R}^{W \times H \times D}$ from the Gaussian normal distribution, such that $x_i \sim \mathcal{N}(0, 1)$ where $x_i$ is its element, and assume that it is located at the server. Since we have $K$ local models $f_{1:K}$, we equally feed the single criteria input $\mathbf{x}_{\mathcal{N}}$ to $K$ local encoders so that we can generate a set of $K$ multiple personalized transformations $\mathcal{Z}_{\mathcal{N}}$ as follows:

$$\mathcal{Z}_{\mathcal{N}} = \{\mathbf{z}_{\mathcal{N},k} | \mathbf{z}_{\mathcal{N},k} = E_k(\mathbf{z}|\mathbf{x}_{\mathcal{N}})\}_{k=1}^K \tag{2}$$

Now our remaining task is to compute the relevancy between the transformations $\mathcal{Z}_{\mathcal{N}}$ which each local knowledge is efficiently encoded. We use the cosine similarity to measure the distances between target model $f_k$ with the rest of models in $\mathcal{F}$, as follows:

$$\Omega(k, \tau, \mathcal{Z}_{\mathcal{N}}, \Theta) = \{(\sigma_i, \theta_i) | \sigma_i = \frac{\mathbf{z}_k \cdot \mathbf{z}_i}{\|\mathbf{z}_k\|\|\mathbf{z}_i\|}, \sigma_i \geq \tau, \mathbf{z}_i \in \mathcal{Z}, \theta_i \in \Theta\}_{i=1}^K, \tag{3}$$

where $\Theta$ is a set of $K$ local weights maintained at the server and $\Omega(\cdot)$ is a similarity function that returns a set of pairs of the similarity score $\sigma_i$ and the corresponding local weights $\theta_i$ for all $i \in \{1, 2, \ldots, K\}$, while satisfying the similarity scores are greater or equal to the given threshold $\tau$.

## 4.2 KERNEL FACTORIZATION

Wang et al. (2020) discussed that the conventional knowledge aggregation, which is performed in a coordinate-wise manner, may have severe detrimental effects on the averaged model. This is because the current deep neural network have often **extremely high dimensional parameter space** and the meaningful element-wise neural matching is not guaranteed when aggregating multiple weights, e.g. $\theta^{(r+1)} \leftarrow \frac{N_k}{N} \sum_{i=1}^{K^{(r)}} \theta_k^{(r)}$.

Such naive aggregation could be less serious when each local updates share common features, for example, in cases of learning from the same data distribution across all clients. However, when each local model learns heterogeneous domains, such simple mixture of knowledge can be highly problematic. It even could lead to *catastrophic forgetting*, which forgets what they have previously learned.

We assume that such problem can be alleviated if the dimensionality of parameter space is significantly reduced as it minimizes information loss and detrimental knowledge collapse. In order to do that, our novel idea is to factorize the high order model parameters into lower rank components, rank 1 vectors. Formally, local model $f_k$ has local weights $\theta_k$ that is a set of weights of its layers, such that $\theta_k = \{\mathbf{w}_k^i\}_{i=1}^L$. The type of each layer $\mathbf{w}_k^i$ then split into fully-connected or convolutional layers.

**Factorization of Fully-Connected Layers**   The dimensionality of the dense weight $\mathbf{w}_k^i$ for the fully-connected layers is drastically increased by multiplication of input and output dimensions, such that $\mathbf{w}_k^i \in \mathbb{R}^{I \times O}$, where $I$ and $O$ indicate respective input and output dimensions. We can reduce the $I \times O$ complexity into rank 1 complexity via factorizing the high order matrix as follows:

$$\mathbf{w}_k^i = \mathbf{u} \times \mathbf{v}, \text{where } \mathbf{u} \in \mathbb{R}^{I \times 1}, \mathbf{v} \in \mathbb{R}^{1 \times O} \tag{4}$$

However, the multiplication of the two vectors interact only in a linear manner, it might be hard to capture non-linearity using just two of them. As such, we add *the light-weighted sparse mask $\boldsymbol{\mu}$* to the weights to additionally capture non-linearity, as follows:

$$\mathbf{w}_k^i = \mathbf{u} \times \mathbf{v} \oplus \boldsymbol{\mu}, \text{where } \mathbf{u} \in \mathbb{R}^{I \times 1}, \mathbf{v} \in \mathbb{R}^{1 \times O}, \boldsymbol{\mu} \in \mathbb{R}^{I \times O} \tag{5}$$

**Factorization of Convolutional Layers**   The difference between fully-connected layers and convolutional layers is that the convolutional layers have multiple kernels (or filters) such that $\mathbf{w}_k^i \in \mathbb{R}^{F \times F \times I \times O}$, where $F$ is a size of filters. Similarly to the factorization of fully-connected layer, we reduce high order parameters into rank 1 vectors and the higly sparse masks to support non-linearity for the convolutional layers, as follows:

$$\mathbf{w}_k^i = \pi(\mathbf{u}' \times \mathbf{v}' \oplus \boldsymbol{\mu}'), \text{where } \mathbf{u}' \in \mathbb{R}^{I \cdot F \times 1}, \mathbf{v}' \in \mathbb{R}^{1 \times O \cdot F}, \boldsymbol{\mu}' \in \mathbb{R}^{I \cdot F \times O \cdot F},$$
$$\pi(\cdot) : \mathbb{R}^{I \cdot F \times O \cdot F} \to \mathbb{R}^{F \times F \times I \times O} \tag{6}$$

Note that we factorize our parameters *before training* and thus the weights of each layer $\mathbf{w}_k^i$ act like the regular weights during training, while learning the basis vectors and the masks separately.

### 4.3   SIMFED ALGORITHM

Putting similarity matching and kernel factorization altogether, we now describe our objective functions for our factorized parameters and personalized knowledge reflection.

**Learning Objective**   Instead of utilizing the single term $\theta_k$ for local weights of neural network $f_k$, now let $\zeta_k, \xi_k$, and $\phi_k$ be sets of $\mathbf{u}_k, \mathbf{v}_k$, and $\boldsymbol{\mu}_k$ of all layers in $f_k$, s.t. $\zeta_k = \{\mathbf{u}_k^i\}_{i=1}^L, \xi_k = \{\mathbf{v}_k^i\}_{i=1}^L$, and $\phi_k = \{\boldsymbol{\mu}_k^i\}_{i=1}^L$, then our local objective function can be defined as follows:

$$\min_{\zeta_k, \xi_k, \phi_k} \sum_{\mathcal{B} \in \mathcal{D}_k} \mathcal{L}(\mathcal{B}; \zeta_k, \xi_k, \phi_k) + \lambda_1 ||\phi_k||_1, \tag{7}$$

where $\mathcal{L}$ is the standard cross-entropy loss performed on all minibatch $\mathcal{B} \in \mathcal{D}_k$. We add the $L_1$ sparsity inducing regularization term to make the mask parameters highly sparse yet to efficiently capture the non-linearity, whose intensity can be controlled by the hyper-parameter $\lambda_1$.

**Personalized Knowledge Reflection**   We avoid direct averaging the knowledge amongst different clients. We instead individually reflect only difference between target local model and the other clients to preserve local reliability more effectively, inspired by Zhang et al. (2021). First, given local model $f_k$, we select the other beneficial knowledge that are relevant to $f_k$ via $\Omega(\cdot)$ described in Eq. 3, returning a set of $J$ pairs of $(\sigma_j, \zeta_j, \xi_j, \phi_j)$. Second, we separately update each factorized parameters $\zeta_j, \xi_j$, and $\phi_j$ while minimizing the collapse of information as follows:

$$\zeta_k \leftarrow \zeta_k + \frac{exp(\alpha \cdot \sigma_j)}{\sum_{1:J} exp(\alpha \cdot \sigma_j)} \cdot \sum_{j=1}^J (\zeta_j - \zeta_k), \tag{8}$$

where $\alpha$ is scalar value for scaling the reflection ratio of similarity score $\sigma_j$. This personalized knowledge reflection is also equally applied to $\xi_j$ and $\phi_j$, respectively (please see Algorithm 1).

---

**Algorithm 1 SimFed Algorithm**

**Function** RunServer()
  initialize $\mathcal{F}, \mathbf{x}_\mathcal{N}, \tau$
  **for** each round $r = 1, 2, \ldots, R$ **do**
    $\mathcal{F}^{(r)} \leftarrow$ select $K^{(r)}$ clients from $\mathcal{M}$
    **for** each client $f_k^{(r)} \in \mathcal{F}^{(r)}$ **in parallel do**
      **if** $r > 1$ **then**
        $(\sigma_{1:J}, \zeta_{1:J}, \xi_{1:J}, \phi_{1:J}) \leftarrow \Omega(k, \tau, \mathcal{Z}_\mathcal{N}, \zeta, \xi, \phi)$
        $\zeta_k^{(r)} \leftarrow \zeta_k^{(r)} + \frac{exp(\alpha \cdot \sigma_j)}{\sum_{1:J} exp(\alpha \cdot \sigma_j)} \cdot \sum_{j=1}^J (\zeta_j^{(r)} - \zeta_k^{(r)})$
        $\xi_k^{(r)} \leftarrow \xi_k^{(r)} + \frac{exp(\alpha \cdot \sigma_j)}{\sum_{1:J} exp(\alpha \cdot \sigma_j)} \cdot \sum_{j=1}^J (\xi_j^{(r)} - \xi_k^{(r)})$
        $\phi_k^{(r)} \leftarrow \phi_k^{(r)} + \frac{exp(\alpha \cdot \sigma_j)}{\sum_{1:J} exp(\alpha \cdot \sigma_j)} \cdot \sum_{j=1}^J (\phi_j^{(r)} - \phi_k^{(r)})$
      **end if**
      $\zeta_k^{(r+1)}, \xi_k^{(r+1)},$
      $\phi_k^{(r+1)}, \mathbf{z}_{\mathcal{N},k} \leftarrow$ RunClient$(\zeta_k^{(r)}, \xi_k^{(r)}, \phi_k^{(r)}, \mathbf{x}_\mathcal{N})$
    **end for**
  **end for**
**Function** RunClient$(\zeta_k, \xi_k, \phi_k, \mathbf{x}_\mathcal{N})$
  **for** each local epoch $e$ from 1 to $E$ **do**
    **for** minibatch $\mathcal{B} \in \mathcal{D}_k$ **do**
      $\theta_{\zeta \times \xi \oplus \phi} \leftarrow \theta_{\zeta \times \xi \oplus \phi} - \eta \nabla \mathcal{L}(\mathcal{B}; \theta_{\zeta \times \xi \oplus \phi})$
    **end for**
  **end for**
  $\mathbf{z}_{\mathcal{N},k} \leftarrow E_k(\mathbf{x}_\mathcal{N}; \theta_{\zeta \times \xi \oplus \phi})$
  **return** $\zeta_k, \xi_k, \phi_k, \mathbf{z}_{\mathcal{N},k}$

---

## 5 EXPERIMENT

### 5.1 EXPERIMENTAL SETUP

**Baseline Models**   We choose our baseline models from three categories, which are the conventional federated learning, personalized federated learning, and the local training model. For the conventional federated learning algorithms, we choose conventionally well-known algorithms, such as FedAvg (McMahan et al., 2017) and FedProx (Li et al., 2018). Particularly, for Agnostic-FL (Mohri et al., 2019), it tackles mutl-domained scenario that is similar to our goals and thus we compare the model with our method. For personalized federated learning methods, we select FedPer (Arivazhagan et al., 2019), Per-FedAvg (Fallah et al., 2020), Clustered-FL (Sattler et al., 2019), and FedFOMO (Zhang et al., 2021). In particular, Clustered-FL clusters clients based on client-level similarity which approach is related to ours. Moreover, FedFOMO can discriminate which clients are helpful or not, which shares a similar approach to our methods. For the local training model, Stand Alone, ironically, it could be the most powerful baseline in such extremely heterogeneous settings, e.g. label- or domain-level heterogeneity, where knowledge transfer becomes non-trivial challenges.

**Training Details**   We use ResNet-9 architecture for all base algorithms and our method and train on $32 \times 32$ sized images. We leverage Stochastic Gradient Descent (SGD) as our optimizer and we ensure that training configurations are equally set for all algorithms, such as 5e-2 for learning rate, 1e-4 for weight decay, 128 for batch size, etc, unless otherwise we state clearly in our paper for a fairness. For our algorithm, we use 0.9 for $\tau$ and 30 for $\alpha$. $\lambda_1$ is set to 1e-4. For all our experiment, please note that the classifier layers are not aggregated in the label- and domain-heterogeneous settings for fairness.

**Datasets**   (1) *Single-domain dataset*: we use three datasets for three respective experiments, which are MNIST, CIFAR-10, and CIFAR-100, for the single-domain scenario. For the first step, we split each dataset into train, validation, and test sets, such that $55,990/7,000/7,000$ for MNIST, $47,985/6,000/6,000$ for CIFAR-10 and 100. We then split the train set into $K$ local partitions $\mathcal{P}_{1:20}$ ($K$=20) in an identical manner (all instances in each class are evenly distributed to all clients) or a non-identical way (instances in each class are sampled from Dirichlet distribution with $\alpha$=0.5). We further permute the labels for each class per local partition $\mathcal{P}_k$ for *Permuted IID* and *Permuted Non-IID* setting to simulate the *Label-Heterogeneity*. (2) *Multi-domain dataset*: we use CIFAR-100 datasets ($60,000$) and split it into 20 datasets $\mathcal{D}_{1:20}$ grouped by 20 superclasses which are also introduced as "coarse" labels, such as mammals, household furniture, vehicles, etc. In each domain $\mathcal{D}_k$ ($3,000$), non-overlapping 5 subclasses are assigned to each client. Then we split each dataset into train ($2,400$), validation($150$), and test ($450$) sets. Please see further details for the dataset configuration in Section 3.2.

### 5.2 EXPERIMENTAL RESULT

**Label Heterogeneity**   As shown in Table 1, for the conventional IID and Non-IID settings, all federated learning methods show higher performance than the pure local training baseline (Stand-Alone), meaning that the locally learned knowledge is helpful each other and sharing them certainly improves each local performance. Interestingly, however, when the labels are not synchronized for each clients (the permuted IID and Non-IID settings), we observed that all existing federated learning methods are degenerated and show even lower than the Stand-Alone models for both IID and Non-IID partitions. Again, note that we do not share the classifier layers for fairness for all algorithms in this permuted settings. The permutation of labels makes your model heterogeneously learn even if the given data is from the same class. Our method, however, shows strongest performance particularly for the permuted settings and actually improves over Stand-Alone models which all other existing models failed. This clearly shows that our method can effectively reflect the other knowledge while minimizing the information collapse via kernel factorization as well as similarity matching.

**Domain Heterogeneity**   Table 2 shows the experimental results for the multi-domain scenario with non-overlapping (disjoint) 20 domains from CIFAR-100 dataset. In this extremely heterogeneous setting, similarly to the permuted labels scenarios, we observe that the prior federated learning algorithms cannot properly share local knowledge amongst clients and fail to achieve the better

Table 1: **Performance Comparison of Single-domain Scenario** We train 20 clients on each dataset for 100 rounds respectively. We measure local accuracy at the final round and report averaged scores over three trials with different seeds. Note that the classifier layers are not aggregated in the permuted settings for fairness.

| Single-domain Federated Learning | MNIST | | CIFAR-10 | | CIFAR-100 | |
|---|---|---|---|---|---|---|
| Method | Conventnl. IID | Permuted IID | Conventnl. IID | Permuted IID | Conventnl. IID | Permuted IID |
| Stand-Alone | 96.91 ($\pm$ 0.13) | 97.26 ($\pm$ 0.06) | 59.62 ($\pm$ 0.41) | 62.01 ($\pm$ 0.27) | 25.12 ($\pm$ 0.24) | 22.92 ($\pm$ 0.12) |
| FedAvg (McMahan et al., 2017) | 97.87 ($\pm$ 0.12) | 95.26 ($\pm$ 0.23) | 65.52 ($\pm$ 0.55) | 45.99 ($\pm$ 0.44) | 45.36 ($\pm$ 0.17) | 11.48 ($\pm$ 0.11) |
| FedProx (Li et al., 2018) | 97.91 ($\pm$ 0.04) | 95.41 ($\pm$ 0.19) | 65.06 ($\pm$ 0.37) | 45.07 ($\pm$ 0.38) | 44.94 ($\pm$ 0.22) | 11.58 ($\pm$ 0.17) |
| Agnostic-FL (Mohri et al., 2019) | 97.87 ($\pm$ 0.10) | 95.13 ($\pm$ 0.13) | 64.69 ($\pm$ 0.32) | 46.37 ($\pm$ 0.71) | 44.63 ($\pm$ 0.21) | 11.73 ($\pm$ 0.12) |
| Clustered-FL (Sattler et al., 2019) | 97.57 ($\pm$ 0.08) | 96.41 ($\pm$ 0.39) | 65.39 ($\pm$ 0.53) | 55.31 ($\pm$ 0.31) | 42.83 ($\pm$ 0.39) | 17.94 ($\pm$ 0.19) |
| FedPer (Arivazhagan et al., 2019) | 98.11 ($\pm$ 0.11) | 94.81 ($\pm$ 0.27) | 68.70 ($\pm$ 0.29) | 45.12 ($\pm$ 0.31) | 49.12 ($\pm$ 0.33) | 11.26 ($\pm$ 0.16) |
| Per-FedAvg (Fallah et al., 2020) | 97.91 ($\pm$ 0.14) | 94.02 ($\pm$ 0.11) | 66.12 ($\pm$ 0.37) | 47.23 ($\pm$ 0.38) | 50.58 ($\pm$ 0.13) | 12.08 ($\pm$ 0.22) |
| FedFOMO (Zhang et al., 2021) | 97.53 ($\pm$ 0.05) | 94.97 ($\pm$ 0.38) | 67.08 ($\pm$ 0.28) | 51.12 ($\pm$ 0.23) | 48.13 ($\pm$ 0.49) | 11.08 ($\pm$ 0.31) |
| SimFed (Ours) | **98.28** ($\pm$ **0.12**) | **97.83** ($\pm$ **0.36**) | **71.01** ($\pm$ **0.19**) | **64.21** ($\pm$ **0.29**) | **52.01** ($\pm$ **0.18**) | **24.11** ($\pm$ **0.23**) |
| **Method** | Non-IID (C) | Non-IID (P) | Non-IID (C) | Non-IID (P) | Non-IID (C) | Non-IID (P) |
| Stand-Alone | 81.52 ($\pm$ 0.43) | 81.73 ($\pm$ 0.45) | 39.77 ($\pm$ 0.11) | 44.46 ($\pm$ 0.22) | 18.53 ($\pm$ xx.xx) | 17.36 ($\pm$ 0.35) |
| FedAvg (McMahan et al., 2017) | 91.89 ($\pm$ 0.47) | 72.34 ($\pm$ 0.19) | 44.37 ($\pm$ 0.31) | 31.43 ($\pm$ 0.15) | 26.32 ($\pm$ 0.34) | 8.87 ($\pm$ 0.36) |
| FedProx (Li et al., 2018) | 89.84 ($\pm$ 0.32) | 66.32 ($\pm$ 0.22) | 42.63 ($\pm$ 0.25) | 31.91 ($\pm$ 0.84) | 26.46 ($\pm$ 0.33) | 5.81 ($\pm$ 0.83) |
| Agnostic-FL (Mohri et al., 2019) | 91.51 ($\pm$ 0.48) | 70.97 ($\pm$ 0.31) | 43.33 ($\pm$ 0.22) | 32.28 ($\pm$ 0.19) | 25.14 ($\pm$ 0.31) | 8.21 ($\pm$ 0.73) |
| Clustered-FL (Sattler et al., 2019) | 90.59 ($\pm$ 0.29) | 77.78 ($\pm$ 0.37) | 49.68 ($\pm$ 0.xx) | 39.33 ($\pm$ 0.34) | 25.73 ($\pm$ 0.xx) | 11.61 ($\pm$ 0.19) |
| FedPer (Arivazhagan et al., 2019) | 90.32 ($\pm$ 0.41) | 70.45 ($\pm$ 0.18) | 50.43 ($\pm$ 0.xx) | 29.45 ($\pm$ 0.53) | 33.58 ($\pm$ 0.xx) | 5.68 ($\pm$ 0.17) |
| Per-FedAvg (Fallah et al., 2020) | 90.66 ($\pm$ 0.35) | 73.37 ($\pm$ 0.43) | 45.91 ($\pm$ 0.73) | 31.23 ($\pm$ 0.62) | 27.39 ($\pm$ 0.86) | 6.49 ($\pm$ 0.33) |
| FedFOMO (Zhang et al., 2021) | 91.74 ($\pm$ 0.33) | 68.94 ($\pm$ 0.29) | 50.62 ($\pm$ 0.93) | 30.15 ($\pm$ 0.35) | 28.29 ($\pm$ 0.23) | 8.93 ($\pm$ 0.19) |
| SimFed (Ours) | **92.13** ($\pm$ **0.27**) | **84.94** ($\pm$ **0.41**) | **52.13** ($\pm$ **0.23**) | **45.37** ($\pm$ **0.23**) | **34.11** ($\pm$ **0.71**) | **18.38** ($\pm$ **0.11**) |

Table 2: **Performance Comparison of Multi-domain Scenario** We train 20 clients on 20 heterogeneous datasets simultaneously for 100 rounds. We measure local accuracy at the final round and report averaged scores over three trials. Note that in the multi-domained settings the classifier layer is not merged for fairness.

| Multi-domain Federated Learning | CIFAR-100, 20 Domains, 5 independent sub-classes for each domain | | | | | | |
|---|---|---|---|---|---|---|---|
| Method | Aqu. Mammals | Fish | Flowers | Food | Fruits & Vege. | Elec. Devices | Furniture |
| Stand-Alone | 50.15 ($\pm$ 2.54) | 62.30 ($\pm$ 1.29) | 64.07 ($\pm$ 5.63) | 68.44 ($\pm$ 0.96) | 73.56 ($\pm$ 2.20) | 67.04 ($\pm$ 2.52) | 65.48 ($\pm$ 1.36) |
| FedAvg (McMahan et al., 2017) | 48.07 ($\pm$ 1.76) | 60.37 ($\pm$ 1.85) | 53.63 ($\pm$ 2.52) | 45.48 ($\pm$ 1.56) | 58.00 ($\pm$ 1.55) | 48.15 ($\pm$ 2.16) | 51.63 ($\pm$ 3.96) |
| FedProx (Li et al., 2018) | 45.63 ($\pm$ 1.00) | 59.48 ($\pm$ 1.36) | 42.96 ($\pm$ 4.79) | 46.07 ($\pm$ 5.31) | 55.41 ($\pm$ 2.45) | 51.78 ($\pm$ 2.02) | 51.93 ($\pm$ 1.79) |
| Agnostic-FL (Mohri et al., 2019) | 48.52 ($\pm$ 1.27) | 59.78 ($\pm$ 1.79) | 52.59 ($\pm$ 0.91) | 46.81 ($\pm$ 3.37) | 54.44 ($\pm$ 3.54) | 49.48 ($\pm$ 1.74) | 51.48 ($\pm$ 3.10) |
| Clustered-FL (Sattler et al., 2019) | 47.78 ($\pm$ 2.40) | 62.81 ($\pm$ 3.63) | 50.81 ($\pm$ 6.14) | 54.89 ($\pm$ 9.47) | 58.07 ($\pm$ 6.08) | 53.33 ($\pm$ 4.63) | 53.19 ($\pm$ 7.64) |
| FedPer (Arivazhagan et al., 2019) | 48.11 ($\pm$ 1.22) | 60.67 ($\pm$ 0.44) | 57.67 ($\pm$ 0.78) | 41.11 ($\pm$ 0.67) | 60.22 ($\pm$ 2.44) | 49.22 ($\pm$ 0.56) | 48.22 ($\pm$ 5.56) |
| Per-FedAvg (Fallah et al., 2020) | 46.78 ($\pm$ 2.11) | 60.00 ($\pm$ 1.11) | 48.33 ($\pm$ 1.67) | 49.78 ($\pm$ 0.00) | 53.00 ($\pm$ 5.44) | 47.44 ($\pm$ 2.33) | 50.89 ($\pm$ 6.67) |
| FedFOMO (Zhang et al., 2021) | 48.07 ($\pm$ 1.36) | 60.00 ($\pm$ 2.27) | 50.67 ($\pm$ 2.09) | 48.15 ($\pm$ 6.23) | 55.26 ($\pm$ 7.42) | 45.56 ($\pm$ 1.45) | 48.44 ($\pm$ 1.89) |
| Similarity Matching (Ours) | 51.56 ($\pm$ 1.45) | 64.67 ($\pm$ 1.89) | 64.52 ($\pm$ 1.00) | 70.67 ($\pm$ 0.63) | 67.41 ($\pm$ 5.87) | 67.78 ($\pm$ 0.65) | **66.96** ($\pm$ **0.69**) |
| + Factorization (Ours) | **56.59** ($\pm$ **1.63**) | **69.11** ($\pm$ **2.75**) | **66.30** ($\pm$ **2.13**) | 70.37 ($\pm$ 1.21) | 73.26 ($\pm$ 1.71) | **72.59** ($\pm$ **1.37**) | 65.93 ($\pm$ 3.52) |
| | Insects | Carnivores | Objects | Outdr. Scenes | Om./Herbivores | Med. Mammals | Invertebrates |
| Stand-Alone | 66.07 ($\pm$ 3.09) | 62.15 ($\pm$ 4.63) | 76.07 ($\pm$ 2.46) | 81.85 ($\pm$ 3.43) | 61.41 ($\pm$ 5.66) | 66.81 ($\pm$ 5.62) | **61.93** ($\pm$ **2.46**) |
| FedAvg (McMahan et al., 2017) | 55.48 ($\pm$ 2.72) | 51.41 ($\pm$ 1.39) | 59.19 ($\pm$ 2.47) | 70.59 ($\pm$ 5.26) | 52.44 ($\pm$ 3.62) | 52.81 ($\pm$ 1.37) | 56.22 ($\pm$ 1.75) |
| FedProx (Li et al., 2018) | 55.26 ($\pm$ 2.15) | 49.04 ($\pm$ 2.72) | 61.04 ($\pm$ 3.54) | 74.89 ($\pm$ 3.15) | 51.41 ($\pm$ 3.29) | 55.04 ($\pm$ 0.86) | 52.59 ($\pm$ 3.82) |
| Agnostic-FL (Mohri et al., 2019) | 54.00 ($\pm$ 0.31) | 49.70 ($\pm$ 4.73) | 57.78 ($\pm$ 6.95) | 73.56 ($\pm$ 1.84) | 50.74 ($\pm$ 3.63) | 54.37 ($\pm$ 3.99) | 49.85 ($\pm$ 2.31) |
| Clustered-FL (Sattler et al., 2019) | 63.56 ($\pm$ 3.00) | 55.41 ($\pm$ 4.03) | 66.59 ($\pm$ 11.63) | 72.07 ($\pm$ 3.67) | 53.70 ($\pm$ 3.12) | 55.70 ($\pm$ 6.09) | 53.41 ($\pm$ 3.75) |
| FedPer (Arivazhagan et al., 2019) | 56.33 ($\pm$ 3.67) | 51.89 ($\pm$ 4.56) | 62.44 ($\pm$ 1.56) | 72.67 ($\pm$ 2.22) | 51.78 ($\pm$ 4.00) | 54.33 ($\pm$ 3.22) | 51.44 ($\pm$ 4.33) |
| Per-FedAvg (Fallah et al., 2020) | 57.89 ($\pm$ 2.56) | 55.22 ($\pm$ 1.00) | 53.78 ($\pm$ 11.78) | 73.44 ($\pm$ 0.33) | 53.56 ($\pm$ 0.89) | 52.44 ($\pm$ 1.11) | 48.89 ($\pm$ 0.44) |
| FedFOMO (Zhang et al., 2021) | 56.37 ($\pm$ 4.81) | 53.85 ($\pm$ 0.73) | 59.63 ($\pm$ 3.09) | 72.00 ($\pm$ 2.09) | 54.00 ($\pm$ 1.84) | 54.37 ($\pm$ 1.82) | 53.41 ($\pm$ 6.52) |
| Similarity Matching (Ours) | 64.81 ($\pm$ 3.98) | 58.67 ($\pm$ 4.91) | 76.44 ($\pm$ 3.21) | **84.44** ($\pm$ **0.36**) | 58.30 ($\pm$ 9.29) | 66.52 ($\pm$ 0.76) | 55.04 ($\pm$ 6.22) |
| + Factorization (Ours) | **69.11** ($\pm$ **3.90**) | **63.56** ($\pm$ **2.67**) | **82.81** ($\pm$ **1.18**) | 83.85 ($\pm$ 0.21) | **66.52** ($\pm$ **4.21**) | **76.15** ($\pm$ **1.41**) | 61.41 ($\pm$ 2.58) |
| | Peopole | Reptiles | Small Mammals | Trees | Vehicles 1 | Vehicles 2 | **AVERAGE** |
| Stand-Alone | 38.74 ($\pm$ 0.46) | 55.78 ($\pm$ 2.27) | 50.22 ($\pm$ 2.09) | 61.19 ($\pm$ 2.46) | **71.48** ($\pm$ **1.65**) | 79.04 ($\pm$ 1.71) | 64.19 ($\pm$ 0.79) |
| FedAvg (McMahan et al., 2017) | 36.07 ($\pm$ 4.46) | 50.00 ($\pm$ 3.19) | 46.37 ($\pm$ 2.42) | 54.96 ($\pm$ 1.48) | 52.22 ($\pm$ 2.18) | 63.85 ($\pm$ 2.16) | 53.35 ($\pm$ 0.76) |
| FedProx (Li et al., 2018) | 34.52 ($\pm$ 1.41) | 49.48 ($\pm$ 1.84) | 44.15 ($\pm$ 1.71) | 52.52 ($\pm$ 2.88) | 48.07 ($\pm$ 1.99) | 63.33 ($\pm$ 0.73) | 52.39 ($\pm$ 0.47) |
| Agnostic-FL (Mohri et al., 2019) | 34.89 ($\pm$ 0.36) | 48.30 ($\pm$ 2.04) | 45.85 ($\pm$ 3.27) | 55.41 ($\pm$ 1.17) | 49.93 ($\pm$ 1.21) | 65.41 ($\pm$ 4.08) | 52.64 ($\pm$ 0.91) |
| Clustered-FL (Sattler et al., 2019) | 37.93 ($\pm$ 1.09) | 53.33 ($\pm$ 1.92) | 47.70 ($\pm$ 1.65) | 56.74 ($\pm$ 0.46) | 51.70 ($\pm$ 3.52) | 66.52 ($\pm$ 3.94) | 55.76 ($\pm$ 2.37) |
| FedPer (Arivazhagan et al., 2019) | 34.33 ($\pm$ 3.67) | 46.44 ($\pm$ 0.44) | 46.33 ($\pm$ 0.78) | 53.78 ($\pm$ 2.00) | 50.33 ($\pm$ 4.11) | 64.44 ($\pm$ 1.78) | 53.09 ($\pm$ 1.56) |
| Per-FedAvg (Fallah et al., 2020) | 38.67 ($\pm$ 0.22) | 52.22 ($\pm$ 0.22) | 46.67 ($\pm$ 0.44) | 52.89 ($\pm$ 4.89) | 53.00 ($\pm$ 1.67) | 61.56 ($\pm$ 5.11) | 52.92 ($\pm$ 1.50) |
| FedFOMO (Zhang et al., 2021) | 36.67 ($\pm$ 1.79) | 52.52 ($\pm$ 0.76) | 40.15 ($\pm$ 1.71) | 53.41 ($\pm$ 0.86) | 52.22 ($\pm$ 1.58) | 61.26 ($\pm$ 6.08) | 53.41 ($\pm$ 6.52) |
| Similarity Matching (Ours) | 36.07 ($\pm$ 1.76) | 58.44 ($\pm$ 1.44) | 50.67 ($\pm$ 1.55) | 59.11 ($\pm$ 1.84) | 68.00 ($\pm$ 5.70) | 76.07 ($\pm$ 2.50) | 63.31 ($\pm$ 0.55) |
| + Factorization (Ours) | **41.26** ($\pm$ **2.89**) | **61.33** ($\pm$ **2.44**) | **52.30** ($\pm$ **4.18**) | **62.00** ($\pm$ **3.36**) | 69.19 ($\pm$ 12.02) | **83.41** ($\pm$ **1.41**) | **67.35** ($\pm$ **0.66**) |

performance over pure local training baseline (Stand-Alone), showing that the domain heterogeneity is critical challenges of federated learning. Even though it seems that federated learning does not work in this domain heterogeneous settings, our method, however, shows the strong possibility. SimFed (Similarity Matching) achieves much higher scores over the existing federated learning algorithms, meaning that it properly utilizes other clients' knowledge. However, it still shows slightly lower

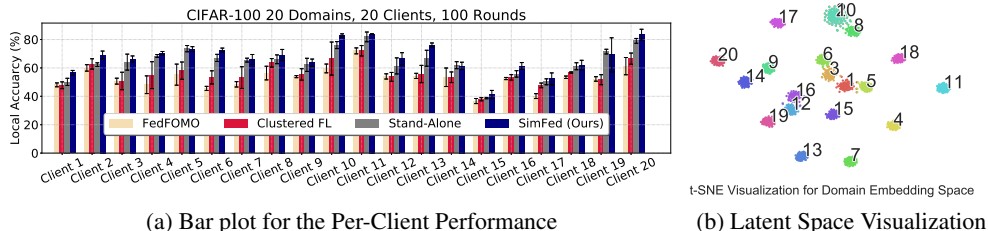

(a) Bar plot for the Per-Client Performance  (b) Latent Space Visualization

Figure 3: **Per-client Performance and Embeddings Space for Local Knowledge:** (a) we show bar plot for the per-client performance comparison. (b) we visualize 2,000 latent embeddings locally learned knowledge (100 Criteria Inputs for 20 Clients). The labeled number is correspondent client id (and also dataset in Table 2)

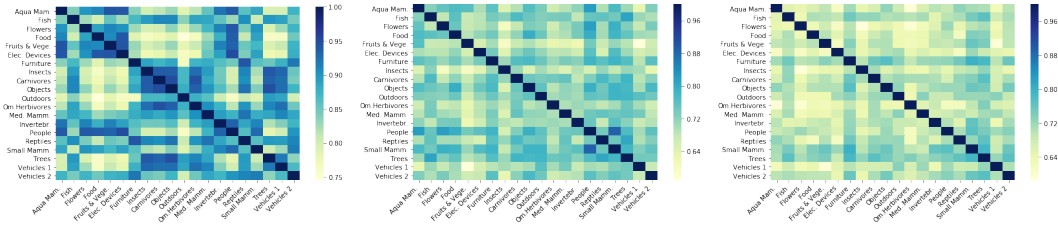

(a) Inter-client Similarity (r=1)  (b) Inter-client Similarity (r=50)  (c) Inter-client Similarity (r=100)

Figure 4: **Inter-Client Similarity:** We visualize inter-client similarity at (a) the beginning with r=1, (b) middle with r=50, and (c) the end with r=100 of 100 training rounds in total.

performance compared to the Stand-Alone model. Even though we choose the beneficial knowledge, the reflection of the knowledge should be properly handled, which is an another critical challenges. With kernel factorization, SimFed finally improves ($4\%p$ higher in average) over Stand-Alone model, meaning that ours can discriminate which knowledge from others are beneficial or harmful and successfully reflects the knowledge to adapt to the given local domain while effectively alleviating the risks caused by domain heterogeneity. In Figure 3 Left, we visualize per-client performance. All clients of our method shows way higher performance over all federated learning methods. In few datasets (Fruits Vege., Invertebrates, and Vehicles 1), ours shows quite marginal performance compared to the Stand-Alone method. Intuitively, we conjecture that those datasets are not really correlated to other local domains in the feature level aspects. Even in this case, however, if some new clients from domains similar to those datasets newly participate in the collaborative learning, then we believe that the performance can be improved as ours can be benefited from the relevant clients.

**Alleviation of Information Loss**  Zhang et al. (2021) discussed aggregating with unhelpful clients leads to significant performance degeneration. In this sense, we match only relevant clients amongst all clients based on our similarity matching algorithm, ensuring that irrelevant knowledge aggregation can be effectively prevented. This reduces information loss and knowledge collapse, as we can see Similarity Matching outperforms most federated learning algorithms in Table 2. However, the conventional knowledge aggregation is performed in extremely high dimensional parameter space in a coordinate-wise manner, which may have severe detrimental effects on the averaged model, as studied by Wang et al. (2020). In this sense, we significantly reduce the dimensionality of model parameters into low rank vectors (with highly sparse masks) by our factorization method, which empirical results further improves performance achieved by similarity matching (+ Factorization in Table 2). The results demonstrate information loss is further largely reduced.

**Effectiveness of Similarity Matching**  Our robustness on label and domain heterogeneity firstly comes from correctly discriminating which knowledge is helpful or not. To see the effectiveness, we compare performance between our similarity matching algorithms and Random Matching & Worst Matching baselines on multi-domain scenario with the exactly same configurations as described above. Random matching baseline randomly pick two arbitrary models to aggregate, while worst matching chooses two the most dissimilar models for knowledge reflection at each round. As shown in Table 3, both random and worst matching models significantly suffer from

Table 3: Similarity Matching

| Method | Acc.(%) |
| --- | --- |
| Worst Matching | 48.83% |
| Random Matching | 52.12% |
| **SimFed (Ours)** | **67.35%** |

the performance degeneration compared to ours (around $15\%p$ for Random Matching and $20\%p$ for Worst Matching), demonstrating that we properly choose the beneficial knowledge for each round.

In Figure 4 we visualize inter-client similarity during training under multi-domain scenario. As shown in the figure, local models seem to be strongly correlated with each other at the first step. However, as learning progressed, we can observe that clients indeed are related to just few of them, which makes more sense in this multi-domain scenario, demonstrates why aggregating all other clients leads to significant performance degeneration. Also, we see that our algorithm can identify the relation between clients more precisely as learning processed. In Figure 3 Right, we visualize our latent embeddings of locally learned knowledge. We utilize 100 Gaussian criteria inputs and transform them by 20 local clients (after 100 rounds training in multi-domain scenario), **yielding** $2,000$ **latent embeddings of local knowledge.** As shown in in the figure, we can observe that the latent embeddings are grouped clearly and does not overlap on the other domains. This shows our encoding method successfully encodes the same things same and the different things differently. This reliability and consistency can help discriminating task-level similarity, which eventually improves local performance.

**Effectiveness of Kernel Factorization**   Even though we successfully match the relevant clients, merging the knowledge should be carefully handled as we observed in multi-domain scenario (Table 2). Minimizing knowledge collapse and information loss is critical in such heterogeneous scenarios. Our kernel factorization method further improves performance achieved by only similarity matching technique as shown in Table 2. This is because we significantly reduce the dimensionality of model parameter space into low rank vector space.

Table 4: Kernel Factorization

| Method | Acc.(%) |
|---|---|
| **SimFed (Ours)** | **67.35%** |
| w/o Factorization | 63.31% |
| w/o Masks $\phi$ | 22.26% |

This alleviates the information loss when merging the heterogeneous knowledge. For further analysis, we compare two baseline models, which are ours with no factorization (w/o Factorization) and ours with no masks (w/o Masks $\phi$), in multi-domain scenario with the same training configurations. When we eliminate the factorization technique (which is equivalent to the Similarity Matching only), the model shows about $4\%p$ lower performance compared to our full model, demonstrating factorization has beneficial effect on improving performance in multi-domain scenario. When we remove the sparse masks $phi$, we observe significant performance drop, demonstrating it successfully captures the non-linearity supporting the low rank vectors.

Table 5: More Factorization Analysis

| Method | IID (P) | NonIID (P) |
|---|---|---|
| FedAvg | 45.99% | 31.43% |
| + Factorization | 49.72% | 34.75% |
| **SimFed (Ours)** | **64.21%** | **45.37%** |

In another analysis, we conversely add our factorization technique to the existing federated learning methods, e.g. FedAvg + Kernel Factorization, and see how the factorization affects the pure federated learning method. We conduct experiments on CIFAR-10 in permuted iid and non-idd settings for 100 rounds. As shown, the combined model shows $3-4\%p$ higher accuracy over pure federated learning algorithms. As the only difference between pure and combined models is the dimensionality of parameter space, which we demonstrate our factorization method alleviate the knowledge collapse and information loss caused by the coordinate-wise aggregation in high dimensional parameter space. Ours still outperforms the baseline model, meaning that our similarity matching algorithms further improves performance.

## 6   CONCLUSION

In this work, we discussed the realistic problem of federated learning at worldwide scenarios, where (1) the labeling schemes are possibly not synchronized across all participants and (2) local clients can be from any multiple domains, which we named such challenges as *Label Heterogeneity* and *Domain Heterogeneity*. To tackle these problems, we proposed a novel method, *Similarity Matching and Kernel Factorization (SimFed)*. Our method measures task-level similarity based on locally learned knowledge and matches the relevant ones for personalized knowledge reflection. Furthermore, we factorize our model parameters into two basis vectors and the highly sparse masks to significantly reduce the dimensionlaity of parameter space for alleviating knowledge collapse and information loss when reflecting the heterogeneous knowledge. We validate our method on both single domain datastes (Permuted IID/Non-IID) and multi-domain datasets (CIFAR-100 20 superclasses) and show our method outperforms the current state-of-the-art approaches on all experiments.

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
