# OpenReview forum: "Agnostic Personalized Federated Learning with Kernel Factorization"
_ICLR.cc/2022/Conference — ICLR 2022 Submitted_

### Official Review · Reviewer_o4KB · 2021-10-30

**Correctness:** 2
**Technical Novelty And Significance:** 2
**Empirical Novelty And Significance:** 2
**Recommendation:** 3
**Confidence:** 4

**Main Review:**

Pros:
(1)	The two scenarios in federated learning studied in this paper including label heterogeneity and domain heterogeneity are interesting and practical.
(2)	The proposed method views the individual local models as respective encoders that can transform their local data distribution to embedding vectors shared among clients, instead of adding any secondary models, which is simple but effective.
Cons:
(1)	In Section 4.2, authors assume that if the dimensionality of parameter space to compute is significantly reduced as well as knowledge is separately captured, then the loss of essential information of model aggregation in domain heterogeneity scenario can be alleviated, which is not well justified.
(2)	There is no reason why the basis vector parameters u can capture the input-related knowledge and v can learn the output-related knowledge.
(3)	There is no detailed justification or no reference to key formulas in the paper. The Eq. (8) for updating factorized parameters is mistaken.
(4)	Some experimental results in Table 1, Table 3 and Table 4 are missing, and the setting of similarity threshold is not presented, which harms the reproducibility of the proposed method. The variant method “w/o Factor” in Table 4 is confusing.
(5)	Given the heterogeneity of different local datasets, the projection these datasets into a latent intermediate space for all clients can not well ensured. The authors did not give sufficient rational analysis for this assumption. This operation may destroy the privacy leak of local data.
The studied problem in this paper is interesting and practical, but the technical idea and technical contents are not well justified, and the experimental results are not well organized and analyzed.

====
Comments after rebuttal:
First of all, the authors make good efforts during the rebuttal period and clarify my concerns on privacy issue. There are still several issues not well addressed.
1. I just checked the author's response and the comments from other reviewer. From my own experience in personalized federated learning, existing solutions indeed can handle different label spaces or non-i.i.d. data across clients. FedBN (ICLR2021) can handle the feature heterogeneity. The authors do not well position their own work and the main technical idea is not well analyzed yet. I admire the addtional results and more ablation study.
2. The authors still do not well revise Eq. (8), the subscript j for the numerator is not well referred from the context, why not merge with the summation also not discusssed.
3. The kernel factorization is not well analyzed yet, mostly are simply supported by improved results.
4. Overall, the main technical idea is not well analyzed and manuscript is not so clearly presented yet. I would like to keep my scorings.

**Summary Of The Paper:**

This paper studies two challenges of personalized federated learning: (1) Label Heterogeneity where label schemes are not synchronized in local clients and (2) Domain Heterogeneity where the datasets owned by the clients can be semantically dissimilar. The authors propose a method called Similarity Matching and Kernel Factorization which measures semantic similarity/dissimilarity between locally learned knowledge and aggregates the relevant ones. Furthermore, the method factorizes the model parameters into two basis vectors and sparse masks to capture representations of the heterogeneous knowledge.

**Summary Of The Review:**

This paper studies the novel and interesting scenarios in federated learning, i.e. label heterogeneity and domain heterogeneity, and the experimental results show that the proposed method outperforms other federated learning methods on single- and multi-domain datasets. The main logical structure of the paper is relatively clear. However, there are also some shortcomings that need to be improved, e.g. the proposed assumption is not well justified and the experimental setting should be detailed. Overall, this manuscript is not mature enough.

---

> ### Author Response · Authors · 2021-11-19
> **Thank you for reviewing our paper. Here are our responses. (Part 2)**
>
>
>
> **Requiring sufficient rational analysis and privacy concerns for dataset projection**
>
> - This is a **critical misunderstanding**. First of all, our method transforms high dimensional model parameters into **vectors** whose outer products reconstruct the original parameters, and this procedure is obviously more secure than handling **raw parameters** directly as most federated learning methods do (transmitting the raw parameters to the server, calculating similarity via gradients, etc)
>
> - We compute the task-level similarities of the local models by using **noise inputs** and not using the **datasets**. This may be the part where the privacy concerns came from, but we do not use any means that allows the input data instances to be reconstructed (e.g. gradients), and thus is as secure as existing methods on federated learning.
>
> - Second, the rationale for such projection is to discriminate beneficiality amongst locally learned knowledge in an efficient and effective way. Discriminating relevant clients is extremely important to improve performance in the heterogeneous settings as discussed in [2]. Prior works, e.g. Clustered-FL or FedGroup, use raw gradient updates to calculate similarity across clients. Further, [2] even downloads other clients' weights and validates locally which one should be merged. The methods are not that computationally efficient. Ours, however, effectively encodes local knowledge and efficiently calculates the similarity with just vector representation.
> We enhanced rationale for both similarity matching (Section 4.1) and kernel factorization (section 4.2) in the new revision.
>
> ---
>
> **Missing or misleading equations, reference, results, and training details**
>
> - In this rebuttal period, we have extensively revised our paper and corrected all missing parts you mentioned. Thank you for careful reading and helping us to revise our paper. All your comments were valuable to update our paper.
>
> - We also described detailed training configurations, including hyper-parameters, in Section 5.1. This can be helpful to reproducing our results.
>
> - In the new revision, the paragraph for “Effectiveness of Kernel Factorization” (10p) is entirely revised. Please go over the revised paper.
>
> ---
>
> [1] Wang et al, Federated Learning with Matched Averaging, ICLR 2020.
>
> [2] Zhang, et al., Personalized Federated Learning with First Order Model Optimization, ICLR 2021

---

> ### Author Response · Authors · 2021-11-19
> **Thank you for reviewing our paper. Here are our responses. (Part 1)**
>
>
> Thank you for reviewing our paper and your constructive suggestions. We extensively revised our paper reflecting all your comments and suggestions. Please go over our responses and check our new revision (updated lines are colored in blue).
>
> ---
>
> **Justification of Section 4.2**
>
> - For the justification of kernel factorization method, we conducted additional experiments and added in-depth analysis in the new revision (Section 5)
>
> - [1] discussed the conventional coordinate-wise aggregation in high dimensional parameter space can be detrimental to the model performance. Our kernel factorization is designed to alleviate such detrimental phenomena by reducing the dimensionality, to perform model aggregation in a more meaningful space. As shown below, **removing the kernel factorization results in significant performance degeneration**.
>
> | CIFAR-100 (Multi-Domain)       | Accuracy (%) |
> |------------------------------- |--------------|
> | **Ours**                       |   **67.35%** |
> | Ours w/o Kernel Factorization  |   63.31 %    |
>
> - In the revised version of the paper, we conversely added kernel factorization technique to the naive federated learning algorithm to see if it improves the performance. We combined it with FedAvg (FedAvg + Kernel Factorization) and conducted experiments on CIFAR-10 for 100 rounds in Permuted IID and Permuted Non-IID settings.
>
> | CIFAR-10  (Single Domain)      | Permuted IID | Permuted NonIID |
> |------------------------------- |--------------|-----------------|
> | Method                         | Accuracy (%) | Accuracy (%)    |
> | FedAvg                         | 45.99%       |     31.43%      |
> | FedAvg + Kernel Factorization  | 49.72%       |     34.75%      |
> | **Ours**                       | **64.21%**   |     **45.37%**  |
>
> - As shown, **kernel factorization does improve the performance of base federated learning algorithms** even without similarity-based matching, which demonstrates that it alleviates knowledge collapse when merging local weights. Since the only difference between pure and combined models is the dimensionality of parameters, we can conclude that reducing the dimensionality helps performance improvement according to the results.
>
> - Besides, Ours still outperforms the baseline model, which further suggests that our similarity matching algorithm further improves performance, by matching local model that are beneficial to each other. We newly added qualitative analysis of similarity matching in the new revision. As shown in the figure 4, we visualize inter-client similarity during training under multi-domain scenarios. As shown in the figure, local models seem to be  strongly correlated with each other at the first step. However, as learning progresses, we can observe that clients evolve to be related to just a few others, which makes more sense in this multi-domain scenario, demonstrating why aggregating all other clients leads to significant performance degeneration. [2] also discussed combining unhelpful clients leads to significant performance degeneration. We efficiently and effectively discriminate which clients are helpful or not and these further improve our performance eventually. Also, we see that our algorithm can identify the relation between clients more precisely as learning processes.
>
> - In the revised version of the paper, we removed all ambiguous expressions related to the basis vector parameters $u$ and $v$ you pointed out. Also, we added the illustration of the overview of our algorithm (Figure 2) to help the reader better understand the high-level concept.
>
> ---

---

> ### Author Response · Authors · 2021-11-26
> **A gentle reminder**
>
> Dear Reviewer o4KB,
>
> We sincerely appreciate your constructive comments. We have made every effort to faithfully address all your comments in the responses. Here, we briefly summarize the main points of our responses below:
>
> - We have completely revised our paper (unclear explanations, missing parts, misleading notations, etc.).
> - We enhanced justification for both Similarity Matching (Section 4.1) and Kernel Factorization (Section 4.2)
> - We conducted additional experiments on Similarity Matching (Section 5.2) and Kernel Factorization (Section 5.2)
> - We added further details (baselines, related works, training configurations) and the high-level illustration for better understanding (Figure 2)
>
> We sincerely appreciate your insightful and constructive comments, and thank you again for your time and efforts in reviewing our paper. Please let us know if you have any further questions.
>
> Best regards, Authors

---

### Official Review · Reviewer_EkKK · 2021-11-02

**Correctness:** 3
**Technical Novelty And Significance:** 3
**Empirical Novelty And Significance:** 3
**Recommendation:** 6
**Confidence:** 3

**Main Review:**

The problem studied in this paper as well as presented experimental results are interesting. However, the paper has some weaknesses:

1- It seems that the paper assumes that models, are neural networks (either with dense layer of convolutional layers). However, this is not mentioned in sections 1 and 3. Furthermore, this paper studies only classification problems. However, authors do not make this clear in the paper especially in section 3. Therefore, the problem setting and assumptions are not clearly defined in the paper. The authors are suggested to revise section 3 to address this issue.

2- The paper needs more rounds of proof-reading and there are some grammatical errors particularly in section 4.1 which makes understating this paper difficult. I think the paper is not well-written and the presentation of the paper in this sense should be improved. For example, in section 4.1, the sentences "Our underlying assumption is that we can discriminate which local models are helpful or harmful each other if we can differentiate which local data distributions are semantically relevant or irrelevant." and "we can measure similarity between the local domains in the embedding space, which eventually enables us to determine which models are beneficial each other accordingly." are not clear that what they want to express. In fact, the authors do not make it clear that what it means by stating the models benefit each other. I think the authors should specify that in what sense the models can be harmful or helpful to each other.

3- There are some personalized federated learning algorithms as well as agnostic federated learning algorithm. I am curious that what happens if we can simply combine these algorithms together to obtain some agnostic personalized federated learning algorithms. It would be great if the authors can compare the performance of their algorithm with some algorithms which are trivial combination of agnostic federated learning and personalized federated learning. This can better shows the effectiveness of the proposed algorithm dealing with label heterogeneity.

**Summary Of The Paper:**

The paper proposes a personalized federated learning algorithm to deal with label heterogeneity and domain heterogeneity. Specifically, the paper focuses on the cases where clients performs classification tasks while the label set of data samples are vary among clients. Furthermore, the data samples may be distributed among clients from different domains.

**Summary Of The Review:**

Overall, I think the paper studies an interesting problem. However, there some issues in the paper that need to be addressed.

---

> ### Author Response · Authors · 2021-11-19
> **Thank you for reviewing our paper. Here are our responses.**
>
>
> Thank you for the valuable comments. We entirely revised our paper reflecting all your suggestions (updated contents are colored in blue). We sincerely appreciate it if you go over our responses and the new revision.
>
> ---
>
> **Revise Section 3 -- clarify that the model means neural networks and the task to solve is image classification problems.**
>
> - In the new revision, we precisely indicate "Our main task is to solve image classification problems using deep neural networks." at the very beginning of Section 3.1. Previously we only mentioned “the C-way multi-class classification problem” in the middle of Section 3.1.
> Also, in the new revision, we replace notations for models ($m$) with neural networks ($f$) throughout the entire paper for more clear description. In the previous draft, both the neural network $f$ and model $m$ are defined, which may cause confusion to readers. We clearly revise the notations and definitions and thank you for your suggestions.
>
> ---
>
> **I think the authors should specify in what sense the models can be harmful or helpful to each other.**
>
> - Indeed, such terminology or keywords are common expressions in (personalized) federated learning particularly tackling local heterogeneity. For example, FedFOMO [1], one of the recent works, also similarly expressed -- "how much a client can benefit from another’s model while restricting unhelpful models".
>
> - We define beneficiality of clients as the degree of performance improvement when merging their knowledge. If merging knowledge between certain clients improves individual model's performance (or global model performance), then those participants can be seen as beneficial to each other. If not, participants can be seen as not helpful to each other, which we should avoid merging them.
>
> - In this sense, the mention in our paper, *“Our underlying assumption is that we can discriminate which local models are helpful or harmful to each other if we can differentiate which local data distributions are semantically relevant or irrelevant”* represents that **we can find the models helpful to each other if we can identify local data similarity**. This assumption is inspired by [1], which revealed that the other clients’ knowledge can affect either negatively or positively to the local performance.
>
> - In the same context, *“We can measure similarity between the local domains in the embedding space, which eventually enables us to determine which models are beneficial to each other accordingly."* also means that **our method can efficiently measure task-level similarity, and this allows us to identify which models are helpful to each other**.
>
> - As suggested, we described "beneficiality between models" at the beginning of Section 4.1 and added more descriptions in the new revision.  We also added a high-level concept figure in the new revision to help better understand our framework.
>
> ---
>
> **Add baseline combining agnostic and personalized federated learning**
>
> - One of our baseline models, Agnostic-FL, is indeed a model that you suggested (naive combination of domain-agnostic and personalized learning approach), as it is combined with FedPer technique, which is a personalized federated learning algorithm. FedPer splits a model into upper layers (feature extraction layers) and prediction layers (classifier layers) and leaves prediction layers locally (not sharing with others).
>
> - In our experiments, not only Agnostic-FL, all conventional federated learning algorithms are combined with FedPer technique **for fair comparison**, as they should learn different classifiers in both label- and domain-heterogeneous settings. Without this technique, all learning algorithms will be significantly degraded in learning proper classifiers.
>
> - We specified in the captions of Table 1 and 2 that we leave prediction layers locally for all federated learning algorithms. In the new revision, we further clearly indicate this in Section 5.1.
>
> ---
>
> [1] Zhang, et al., Personalized Federated Learning with First Order Model Optimization, ICLR 2021

---

### Official Review · Reviewer_3krS · 2021-11-03

**Correctness:** 3
**Technical Novelty And Significance:** 3
**Empirical Novelty And Significance:** 2
**Recommendation:** 5
**Confidence:** 4

**Main Review:**

Strengths:
+ The research problem that benefits clients by sharing some common knowledge with a well-designed mechanism is an interesting topic
+ To decompose parameters to lower level rank for local models, to my best knowledge, is a novel idea to explore.
+ The experiment results are sufficient enough to support their claims

Weakness:
- There are some research works discussing how to use general knowledge across clients by dividing parameters into different parts, like continual learning in FL, where each client has several tasks to conduct [1]. Also, the basic idea to group users based on metric similarity has been explored in FL as well [2,3];
- I am a little concerned about the privacy guarantees and extra computational requirements on the server-side. The extracted knowledge from the clients is leveraged by reflecting the difference between the target local model and the other clients. One of the benefits of using FL is protecting users’ privacy. Though clients are not sharing data in this work, we need to pay attention to how to leverage extracted knowledge. Also, the kernel factorization of FC layers and convolutional layers happens on the server-side. I am not sure how much extra computation it would take and whether it would cause a delay in the iterative FL update process.
- More discussion about the effectiveness of similarity matching and kernel factorization is expected. Furthermore, in most FL work, convergence guarantee is also another point that is expected to discuss besides the experiments.

[1] Yoon, Jaehong, Wonyong Jeong, Giwoong Lee, Eunho Yang, and Sung Ju Hwang. "Federated continual learning with weighted inter-client transfer." In International Conference on Machine Learning, pp. 12073-12086. PMLR, 2021.

[2] Duan, Moming, et al. "FedGroup: Efficient Federated Learning via Decomposed Similarity-Based Clustering."

[3] Jeong, Wonyong, et al. "Federated Semi-Supervised Learning with Inter-Client Consistency & Disjoint Learning." arXiv preprint arXiv:2006.12097 (2020).


**Summary Of The Paper:**

Authors propose an algorithm for clients to learn from each other even under the setting of label heterogeneity and domain heterogeneity. The basic idea is they leverage metrics to measure the similarity between the representative knowledge and corporate clients accordingly, which can maintain the generally shared information and reduce information loss at the same time. They conduct appropriate experiments to test the algorithm and provide a comparison with current the-state-of-the-art fl methods. The main contribution is to propose an approach to support clients to benefit each other across different types of labels and domains. They leverage the kernel factorization approach on FC and convolutional layers and update them separately on the server-side. They provide extensive experiment results to support their claims.

**Summary Of The Review:**

The authors do provide a fine solution to solving a real problem of generalizing knowledge across clients by kernel factorization. Extensive experiments are provided as well. However, if we consider each part of the techniques, like model parameter kernel factorization, grouping users based on similarity, I may expect more novel contributions.

---

> ### Author Response · Authors · 2021-11-19
> **Thank you for reviewing our paper. Here are our responses. (Part 2)**
>
>
> ---
>
> **More discussion and analysis for the effectiveness of the proposed methods**
>
> - As advised, we added new analysis for both kernel factorization and similarity matching techniques in the new revision (Section 5.2).
>
> - For the factorization method, we previously showed its effect by removing the kernel factorization. We observed that removing it causes significant performance degradation, as shown below.
>
> | CIFAR-100 (Multi-Domain)       | Accuracy (%) |
> |------------------------------- |--------------|
> | **Ours**                       |   **67.35%** |
> | Ours w/o Kernel Factorization  |   63.31 %    |
>
> - Now, for further analysis, we added our factorization technique to the existing federated learning methods (FedAvg + Kernel Factorization) and see how the factorization affects FedAvg. We conduct experiments on CIFAR-10 in permuted iid and non-iid settings for 100 rounds.
>
> | CIFAR-10                       | Permuted IID | Permuted NonIID |
> |------------------------------- |--------------|-----------------|
> | Method                         | Accuracy (%) | Accuracy (%)    |
> | FedAvg                         | 45.99%       |     31.43%      |
> | FedAvg + Kernel Factorization  | 49.72%       |     34.75%      |
> | **Ours**                       | **64.21%**   |     **45.37%**  |
>
> - As shown, the combined models are improved over pure federated learning algorithms, demonstrating that kernel factorization alleviates knowledge collapse when merging local weights. The difference between pure and combined models is the dimensionality of parameter space in that ours significantly reduces high dimensional parameters into low rank vector space. This observation is also supported by [4] which discussed that coordinate-wise aggregation can be detrimental.
>
> - Ours still outperforms the baseline model, meaning that our similarity matching algorithms further improves performance, while matching clients that are beneficial to each other. We newly added qualitative analysis of similarity matching in the new revision. As shown in the figure 4, we visualize inter-client similarity during training under multi-domain scenarios. As shown in the figure, local models seem to be  strongly correlated with each other at the first step. However, as learning progresses, we can observe that clients indeed are related to just a few of them, which makes more sense in this multi-domain scenario, demonstrating why aggregating all other clients leads to significant performance degeneration. [5] also discussed combining unhelpful clients leads to significant performance degeneration. We efficiently and effectively discriminate which clients are helpful or not and these further improve our performance eventually. Also, we see that our algorithm can identify the relation between clients more precisely as learning processes.
>
>
> ---
> **I may expect more novel contributions.**
>
> - Please consider that our method highly differs from the existing methods in two aspects.
>
> - As described above, ours factorizes high dimensional parameters into low rank vector space **to reduce the dimensionality of parameter space** for alleviating the knowledge collapse, while prior works additively decomposed model parameters to learn by different objectives which do not aim to reduce the dimensionality. Ours can effectively alleviate the knowledge collapse cause by the coordinate-wise aggregation discussed in [4] and this has significant novelty differentiated from the existing decomposition methods.
>
> - Our method also **does not utilize gradient updates directly** to calculate similarity between clients. On the other hand, most existing methods calculate the raw gradients for discriminating relevant clients . Rather, ours uses efficiently and effectively encoded model representations to measure client-level relevance and this is even more secure than the prior methods since it uses much less amount of information without destroying any privacy constraints.
>
> ---
>
> [1] Yoon, et al. Federated Continual Learning with Weighted Inter-Client Transfer, ICML 2021.
>
> [2] Jeong, et a., Federated Semi-Supervised Learning with Inter-Client Consistency & Disjoint Learning, ICLR 2021.
>
> [3] Duan, et al. FedGroup: Efficient Federated Learning via Decomposed Similarity-Based Clustering. ISPA 2021.
>
> [4] Wang et al, Federated Learning with Matched Averaging, ICLR 2020.
>
> [5] Zhang, et al., Personalized Federated Learning with First Order Model Optimization, ICLR 2021

---

> ### Author Response · Authors · 2021-11-19
> **Thank you for reviewing our paper. Here are our responses. (Part 1)**
>
>
> Thank you for your helpful suggestions. We reflected all your valuable comments in our new revision. Please check the revised version of our paper, and check our responses below.
>
> ---
>
> **Prior works that share similar approaches with ours**
>
> - We included in-depth discussions of [1,2,3] in relation to ours in the revision, as you suggested. Thank you for recommending the valuable prior works.
>
> - Our parameter factorization method completely differs from the parameter decomposition techniques proposed by [1] and [3]. We are well-aware of these two works, but did not discuss them since they are only distantly related to our work. Please note that the "parameter decomposition" means two completely different things in those works and ours, since they use **additive parameter decomposition**, factorizing a matrix into a sum of a dense matrix and a sparse matrix, while we factorize the parameters as the **outer product of two vectors**. These are two completely different factorizations.
>
> - The main difference is that ours is designed to **reduce the dimensionality of parameters by projecting them onto the low-rank vector space**. None of the two prior techniques reduces the dimensionality. They create one more additional set of parameters with the same dimensionality **to train with different objectives**, then additively combine the two sets of parameters at inference times. On the other hand, ours **factorizes** given single set of parameters into two **vectors (with sparse masks)** **for reducing the dimensionality** to alleviate the knowledge collapse caused by the coordinate-wise aggregation in high dimensional space which is discussed by [4].
>
> - The major difference of our work from **FedGroup [2]**, is the way of measuring the similarity. While FedGroup uses the gradient updates for calculating similarity, ours uses a single vector embedding obtained by an unbiased criteria input, which is simpler and more efficient to calculate, as well as is more secure. The aggregation strategies are also different. While FedGroup performs intra & inter group aggregation after grouping the clients, ours does not group the clients but dynamically perform individual similarity matching at each round, which is more flexible and does not result in clear groupings. Note that FedGroup works similarly to our Clustered-FL baseline we use in our experiments.
>
> ---
>
> **Concerns about privacy guarantees and server-side computation**
>
> - This seems like a critical misunderstanding since our method **does not yield increased privacy concerns or server-side computations**.
>
> - Please note that our model only sends out the vectors that result from factorizing the weight matrix, and thus contains **less information** compared to the raw weights communicated by existing federated learning algorithms. We also do not use any real data when computing the similarities of the models.
>
> - It seems that you have brought up the privacy concerns since prior works such as FedGroup or Clustered-FL, send the full gradient updates to the server to calculate the similarity. Note that our algorithm **does not send out the gradients or the data**.
>
> - Our kernel factorization also does not affect server-side computation. Firstly, the kernel factorization occurs **at the local clients**, which does not require any extra computations at the server and is also cheap to compute even at the local clients. Secondly, merging the factorized parameters (rank 1 vectors) at the server requires **reduced computational costs**, compared to the merging of high-dimensional parameters, as done by existing federated learning methods.

---

> ### Author Response · Authors · 2021-11-26
> **A gentle reminder**
>
> Dear Reviewer 3krS,
>
> We sincerely appreciate your constructive comments. We have made every effort to faithfully address all your comments in the responses. Here, we briefly summarize the main points of our responses below:
>
> - We have completely revised our paper (unclear explanations, missing parts, misleading notations, etc.).
> - We enhanced justification for both Similarity Matching (Section 4.1) and Kernel Factorization (Section 4.2)
> - We conducted additional experiments on Similarity Matching (Section 5.2) and Kernel Factorization (Section 5.2)
> - We added further details (baselines, related works, training configurations) and the high-level illustration for better understanding (Figure 2)
>
> We sincerely appreciate your insightful and constructive comments, and thank you again for your time and efforts in reviewing our paper. Please let us know if you have any further questions.
>
> Best regards, Authors

---

### Official Review · Reviewer_85V7 · 2021-11-03

**Correctness:** 3
**Technical Novelty And Significance:** 1
**Empirical Novelty And Significance:** 1
**Recommendation:** 3
**Confidence:** 4

**Main Review:**

I have the following major concerns.

- The comparison baselines for Label Heterogeneity do not seem informative. By design, many of these baseline methods would implicitly require the labels to be correctly aligned. A more reasonable way is to compare the proposed framework to those methods that take label mismatch into account. For example, you may simply integrate permutation-robust techniques into FedAvg to deal with label heterogeneity cases. I would like to see more comparisons based on such kinds of baselines.

- It is unclear why we should use the cosine similarity. Each local model may have different preferences of scaling resulting given different domains of the data source. Also, what exactly/mathematically does the keyword ’semantic’ mean?

**Summary Of The Paper:**

This paper investigates the FL under the label and the domain heterogeneity.
Specifically, it considered two settings where (i) the labels may not be precisely aligned (ii) the data domain may not be the same. To tackle these two challenges, it proposed the SIMFED framework which first calculates the similarity between the embedding of different local models and then conducts a kernel factorization to solve the problem of heterogeneous data domains.

**Summary Of The Review:**

Overall, I found the paper is a heuristic work on personalized federated learning with little justification and little motivation. The scope of its application is unclear. The empirical results do not seem to be sufficient to support the claimed contributions.

---

> ### Author Response · Authors · 2021-11-19
> **Thank you for reviewing our paper. Here are our responses.**
>
>
> Thank you for reviewing our paper. We have extensively revised our paper reflecting your comments, and would deeply appreciate it if you would go over the revised paper. Below are our responses to your comments.
>
> ---
>
> **Compare the proposed framework to those methods that take label mismatch into account.**
>
> - To the best of our knowledge, label mismatch is often handled in **federated semi-supervised learning scenarios**, which is beyond the scope of our paper. In the standard supervised (personalized) federated learning scenarios, the label heterogeneity problem coming from considering different label permutations or tasks have not yet been studied. Most existing FL methods only tackle **distributional shifts**, where the data distribution of the training examples across different clients are highly imbalanced (Non-IID). Hence, to our knowledge, there exists no proper technique for handling such label heterogeneity in the standard federated learning settings, and we would appreciate if you could point out a specific method that can handle such a scenario.
>
> - Nevertheless, one thing we can do is to integrate our factorization technique with the existing methods and analyze how the combined model works. To do that, we first combine our factorization method with FedAvg, namely FedAvg + Factorization. Then we conduct experiments on CIFAR-10 for 100 rounds.
>
> | CIFAR-10                       | Permuted IID | Permuted NonIID |
> |------------------------------- |--------------|-----------------|
> | Method                         | Accuracy (%) | Accuracy (%)    |
> | FedAvg                         | 45.99%       |     31.43%      |
> | FedAvg + Kernel Factorization  | 49.72%       |     34.75%      |
> | **Ours**                       | **64.21%**   |     **45.37%**  |
>
> - As shown in the above table, the combined model, FedAvg + KF, improves upon FedAvg, which shows that kernel factorization alleviates knowledge collapse when merging local weights. This performance gain mostly comes from working on a low-rank vector space, and this observation is also supported by [1] which discussed that coordinate-wise aggregation can be detrimental.
>
> - Ours still outperforms the baseline which uses KF alone, which suggests that our similarity matching algorithm further improves allows matching clients that are beneficial to each other. We newly added qualitative analysis of similarity matching in the new revision. As shown in the figure 4, we visualize inter-client similarity during training under multi-domain scenarios. As shown in the figure, local models seem to be  strongly correlated with each other at the first step. However, as learning progresses, we can observe that they diverge as training goes on, and each local model becomes related to only a few other local models. This makes more sense in this multi-domain scenario, demonstrating why aggregating all other clients leads to significant performance degeneration. This is also consistent with the observation in [2], which also discussed that combining unhelpful clients leads to significant performance degeneration. We efficiently and effectively discriminate which clients are helpful or not and these further improve our performance eventually. Also, we see that our algorithm can identify the relation between clients more precisely as learning processes.
>
> ---
> **It is unclear why we should use the cosine similarity.**
>
> - To measure similarity across different local models, we use the model outputs obtained by **a single unbiased criteria input**, which is fed into all models and then yields vector representations with varying degree of similarities from each other. Our hypothesis is that the models trained on similar tasks will have similar representations, and this is what we aim to capture by measuring the cosine similarities across the models.
>
> - While we can use any metrics for measuring the similarities between two vectors that represent the knowledge captured at different local models, since "each local model may have different preferences of scaling resulting given different domains of the data source", as you mentioned, we use cosine similarity to capture the task-level similarity only, rather than scale differences resulting from the scales of the input data.
>
>
> ---
>
> **What exactly/mathematically does the keyword ’semantic’ mean?**
>
> - "Semantic similarity" means "task-level similarity", which tells how similar the true functions mapping inputs to labels are (e.g. classification of food vs. animals). Thank you for pointing out the ambiguous expression. We replaced semantic similarity with task-level similarity in our new revision.
>
> ---
>
> [1] Wang et al, Federated Learning with Matched Averaging, ICLR 2020.
>
> [2] Zhang, et al., Personalized Federated Learning with First Order Model Optimization, ICLR 2021

---

> > ### Comment · Reviewer_85V7 · 2021-11-26
> > **Further response**
> >
> > The keyword "semantic" or "task-level similarity" is still unclear to me.
> > I appreciate the authors' time put into responses. But still, I tend to evaluate the paper as a heuristic work with little justification and fairly weak empirical results.

---

> ### Author Response · Authors · 2021-11-22
> **Additional experimental results**
>
> Dear 85v7,
>
> As you suggested, we conducted an additional experiment to show the effect of the different distance metrics (e.g. Euclidean distance) when matching locally learned knowledge amongst client. We experimented with the same configurations with Table 2(CIFAR-100, 20 clients, 100 rounds, multi-domain scenario). We report averaged performance after three different trials.
>
> | Metric            | Averaged Accuracy (%) |
> |-------------------|-----------------------|
> | Euclidean Distance    | 66.53% ±0.67          |
> | **Cosine Similarity (Ours)** | **67.35% ±0.66**          |
>
> As shown, using Euclidean distant shows slightly lower accuracy (about 1%p ) compared to cosine similarity. This is because, in the multi-domain scenario, which multiple clients learn heterogeneous data, the scale of locally learned knowledge (Euclidean distance) is indeed relatively less informative than the directions of locally learned knowledge (cosine similarity). We conjecture that, even with the heterogeneous scales, if the directions are similar, then those learned knowledge may interact positively.
>
> Thank you for your constructive suggestion, and we will include this result and analysis in our final revision.

---

> > ### Comment · Reviewer_85V7 · 2021-11-26
> > **Followup of the experimental results**
> >
> > The results show that the two distance metrics have no significant difference. The mean difference is comparable to the standard error.

---

> ### Author Response · Authors · 2021-11-26
> **A gentle reminder**
>
> Dear Reviewer 85V7,
>
> We sincerely appreciate your constructive comments. We have made every effort to faithfully address all your comments in the responses. Here, we briefly summarize the main points of our responses below:
>
> - We have completely revised our paper (unclear explanations, missing parts, misleading notations, etc.).
> - We enhanced justification for both Similarity Matching (Section 4.1) and Kernel Factorization (Section 4.2)
> - We conducted additional experiments on Similarity Matching (Section 5.2) and Kernel Factorization (Section 5.2)
> - We added further details (baselines, related works, training configurations) and the high-level illustration for better understanding (Figure 2)
>
> We sincerely appreciate your insightful and constructive comments, and thank you again for your time and efforts in reviewing our paper. Please let us know if you have any further questions.
>
> Best regards,
> Authors

---

### Official Review · Reviewer_9bEM · 2021-11-04

**Correctness:** 3
**Technical Novelty And Significance:** 3
**Empirical Novelty And Significance:** 2
**Recommendation:** 6
**Confidence:** 3

**Main Review:**

Strengths:
Well motivated challenges to tackle: label heterogeneity and domain heterogeneity. Then the authors propose similarity matching and kernel factorization to address the issues and show compelling results in experiments.

Weakness:
1. In table 2, it shows that Similarity Matching + Factorization did not outperform Stand-Alone for the "Fruits & Vege" dataset. It would benefit if authors could clarify why, and share some intuitive explanation why or in what scenarios that the proposed algorithms underperform comparing to the "stand alone" approach.
2. The paper would benefit with some explanation why each baseline models & architecture were chosen here for comparison.
3. It was not clear why SimFed reduces information loss when reflecting the heterogeneous knowledge.
4. Kernel Factorization outperform SimFam in most datasets, excepts some. It would benefit if the paper could explain why certain dataset behaves that.

Minor typo:
1. In page 2, it says "We exhaustively validate our method on both singl[e] domain datastes (Permuted IID/Non-IID) and multi domain datasets (CIFAR-100 20 superclasses) and show our method outperforms the current state-of-the-art approaches on all experiments." It should be "single" rather than "singl".


**Summary Of The Paper:**

The main contribution of the work are as follow:
1. introduce the problem of Agnostic Personalized Federated Learning (APFL), and discuss its two possible issues (Label- and Domain-Heterogeneity)
2. propose the method called Similarity Matching and Kernel Factorization (SimFed) to tackle the problem.
3. Validate the method in both label- and domain-heterogeneous scenarios, and show its superiority.

**Summary Of The Review:**

marginally above the acceptance threshold, given that the weakness shared above. Overall it was well motivated problem, and the authors proposed approaches and show compelling results. As pointed out in the above for its weakness, the paper would be strong by providing these insights and intuitions.

---

> ### Author Response · Authors · 2021-11-19
> **Thank you for reviewing our paper. Here are our responses.**
>
>
> Thank you for your constructive comments. We carefully responded to all your comments. We also reflected your valuable suggestions in our new revision. We would appreciate it if you check the new revision.
>
> ---
>
> **In Table 2, why did the proposed method underperform the Stand-Alone method on some sub-datasets (e.g. Fruits & Vege.)?**
>
> - While our method largely outperforms all existing (personalized) federated learning models, as you mentioned, it shows quite marginal performance compared to the Stand-Alone method on some datasets (Fruits & Vege., Invertebrates, and Vehicles 1). Intuitively, we conjecture that those datasets are not really correlated to other local domains in the feature level aspects.
>
> - We can overcome such a phenomenon if some clients newly participate in the learning phase, which are from similar domains to those datasets. They will share their relevant knowledge and benefit each other. This is our strong point, since our learning framework can get better and better as the participation and application scale is getting larger.
>
> ---
> **Choice of baseline models**
>
> - We choose our baseline models based from **three categories**, which are (1) the conventional federated learning, (2) personalized federated learning, and (3) local training algorithm.
>
> - For the  conventional federated learning algorithms, we choose conventionally well-known algorithms, such as **FedAvg** and **FedProx**. Particularly, for **Agnostic-FL**, it tackles mutl-domained scenario (by introducing regularization terms) that is similar to our goals and thus we compare our method to it.
>
>  - For personalized federated learning methods, we select **FedPer**, **Per-FedAvg**, **Clustered-FL**, and **FedFOMO**. In particular, **Clustered-FL** clusters clients based on client-level similarity which approach is related to ours and thus we compare ours with it. Moreover, **FedFOMO** is the current state-of-the-art personalized federated learning method and can discriminate which clients are helpful or not, which shares a similar approach to our methods and thus we compare ours with it as well.
>
> - For the local training model, **Stand Alone**, ironically, it could be the most powerful baseline in such extremely heterogeneous settings, e.g. label- or domain-level heterogeneity, where knowledge transfer becomes non-trivial challenges. Hence we compare our method with this local training model to show knowledge transfer is still possible in such scenarios.
>
> ---
>
> **Why does the proposed method reduce the information loss?**
>
>
> - First, conventional knowledge aggregation is performed in extremely high dimensional parameter space in a coordinate-wise manner, which may have severe detrimental effects on the averaged model, which is discussed in [1]. Rather, we significantly reduce the dimensionality of parameter space into low rank vector space (with additional light-weighted sparse masks) by our factorization method, which empirical results show significant performance improvement, demonstrating information loss is largely reduced.
>
> - Second, [2] discussed aggregating irrelevant knowledge cannot be helpful to local models’ performance. We match only relevant clients amongst all clients based on our similarity matching algorithm, ensuring that irrelevant knowledge aggregation can be effectively prevented. This further reduces information loss and knowledge collapse.
>
> - Lastly, for further reducing information loss, we avoid naively aggregating locally learned knowledge but rather reflecting the difference of weights between local weights and the other clients’ weights. We then scale the gap by degree of similarity to dynamically adjust the ratio of reflection.
>
>
> ---
>
>
> **Why does kernel factorization further improve performance?**
>
>
> - Even though we can discriminate which knowledge is helpful or not, reflecting relevant (but still not identical) knowledge should be carefully handled. As [1] revealed that the conventional coordinate-wise aggregation can be very harmful for the averaged model, we should avoid such naive aggregation in extremely high dimensional parameter space.
>
> - As described in the above response, our kernel factorization method significantly reduces high dimensional parameters into low rank vectors, which largely alleviates the information loss when merging the locally learned knowledge. This further improves performance over similarity matching only and the Stand-Alone models.
> ---
>
> [1] Wang et al, Federated Learning with Matched Averaging, ICLR 2020
>
> [2] Zhang, et al., Personalized Federated Learning with First Order Model Optimization, ICLR 2021

---

### Author Response · Authors · 2021-11-19
**Thank you for your efforts and constructive comments**

Dear all reviewers,

Thank you all for your constructive comments and extensive efforts for reviewing our paper. We carefully read all comments and exhaustively answered all of them. Moreover, we have **fully revised our paper, faithfully reflecting all your comments and suggestions**. We believe that your valuable suggestions have significantly strengthened our paper. Please go over the revised paper as well as our responses to your comments below.

Thank you,
Authors

---

### Author Response · Authors · 2021-11-22
**A gentle reminder**

Dear Reviewers,

We would like to gently remind you that our discussion period will end Today. We have faithfully addressed your concerns and conducted several experiments for justification of our method. Also, we have completely revised our paper reflecting your valuable suggestions. The following is the quick summary of the response.

- We have completely revised our paper (unclear explanations, missing parts, misleading notations, etc.).
- We enhanced justification for both Similarity Matching (Section 4.1) and Kernel Factorization (Section 4.2)
- We conducted additional experiments on Similarity Matching (Section 5.2 Response to 85V7) and Kernel Factorization (Section 5.2)
- We added further details (baselines, related works, training configurations) and the high-level illustration for better understanding (Figure 2)

Please let us know if there is anything else you want us to clarify. We thank you again for your constructive comments.

Best regard,
Authors

---

### Decision · Program_Chairs · 2022-01-20

**Decision:**

Reject

**Comment:**

The paper studies two aspects of personalized federated learning: (1) Clients having their own labeling scheme. (2) Domain heterogeneity across clients. They propose a way to collaborate across clients by similarity matching. The key novelty is to measure similarity of client pairs, based on on how much their representation layer agrees (measured with cosine similarity). A second novelty is a low-rank factorization of model weights. Empirical evaluations show wins on MNIST, CIFAR10, 100.

Reviewers had various grave concerns. On the method side, they were concerned that thee is not enough theoretical insight and analysis of the proposed approach, esp. the kernel factorization and its effect.  On the empirical side, they were concerned that comparisons were not made with most recent baselines. There was a large number of PFL approaches published in 2021. e.g. FedBN.  Among these, its worth noting pFedHN (ICML2021) which actually discussed the case of heterogeneous (permuted) labels (their Sec 3.3).

In a discussion, reviewers appreciated the responses by the authors, the additional experiments and ablation studies. Unfortunately however, they found that the paper is not ready for publication in ICLR.